# LARGE-VOCABULARY 3D DIFFUSION MODEL WITH TRANSFORMER

**Ziang Cao[1], Fangzhou Hong[1], Tong Wu[2,3], Liang Pan[1,3], Ziwei Liu[1]***
[1]S-Lab, Nanyang Technological University, [2]The Chinese University of Hong Kong,
[3]Shanghai Artificial Intelligence Laboratory
`{ziang.cao,fangzhou.hong,liang.pan,ziwei.liu}@ntu.edu.sg`
`wt020@ie.cuhk.edu.hk`

## ABSTRACT

Creating diverse and high-quality 3D assets with an automatic generative model is highly desirable. Despite extensive efforts on 3D generation, most existing works focus on the generation of a single category or a few categories. In this paper, we introduce a diffusion-based feed-forward framework for synthesizing massive categories of real-world 3D objects *with a single generative model*. Notably, there are three major challenges for this large-vocabulary 3D generation: **a**) the need for expressive yet efficient 3D representation; **b**) large diversity in geometry and texture across categories; **c**) complexity in the appearances of real-world objects. To this end, we propose a novel triplane-based 3D-aware **Diff**usion model with **T**rans**F**ormer, **DiffTF**, for handling challenges via three aspects. **1**) Considering efficiency and robustness, we adopt a revised triplane representation and improve the fitting speed and accuracy. **2**) To handle the drastic variations in geometry and texture, we regard the features of all 3D objects as a combination of generalized 3D knowledge and specialized 3D features. To extract generalized 3D knowledge from diverse categories, we propose a novel 3D-aware transformer with shared cross-plane attention. It learns the cross-plane relations across different planes and aggregates the generalized 3D knowledge with specialized 3D features. **3**) In addition, we devise the 3D-aware encoder/decoder to enhance the generalized 3D knowledge in the encoded triplanes for handling categories with complex appearances. Extensive experiments on ShapeNet and OmniObject3D (over 200 diverse real-world categories) convincingly demonstrate that a single DiffTF model achieves state-of-the-art large-vocabulary 3D object generation performance with large diversity, rich semantics, and high quality. More results are available at `https://difftf.github.io/`.

## 1 INTRODUCTION

Creating diverse and high-quality 3D content has garnered increasing attention recently, which could benefit many applications, such as gaming, robotics, and architecture. Recently, various advanced techniques have achieved promising results in 3D generation (Sitzmann et al., 2019; Müller et al., 2022; Achlioptas et al., 2018; Nichol et al., 2022; Chan et al., 2022; Mo et al., 2023). However, many of these methods excel in a specific category but struggle to maintain robustness across a wide range of objects. Consequently, there is an urgent need to develop an approach capable of generating high-quality 3D objects across a broad vocabulary.

Compared with the prior works on single-category generation, large-vocabulary objects have three special challenges: a) the urgent requirement for expressive yet efficient 3D representation; b) wide diversity across massive categories; c) the complicated appearance of real-world objects. Since the explicit ways are easy to evaluate but compute-intensive and implicit ways are easy to extend but time-consuming in evaluating, we adopt an efficient hybrid 3D representation, *i.e.*, Triplane feature (Chan et al., 2022). Besides, benefiting from state-of-the-art (SOTA) performance and reasonable denoising process, diffusion-based methods have attracted much attention in 3D object generation (Müller et al.,

---

*Corresponding author

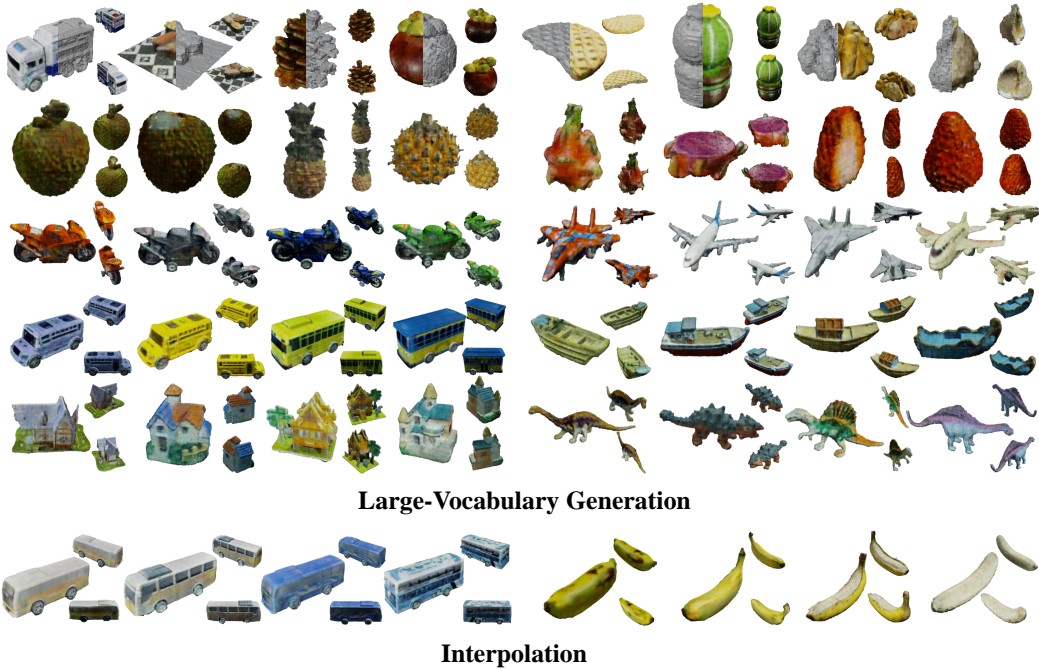

**Large-Vocabulary Generation**

**Interpolation**

Figure 1: **Visualization on Large-vocabulary 3D object generation.** DiffTF can generate *high-quality* 3D objects with *rich semantic information* and *photo-realistic RGB*. Top: Visualization of the generated results. Bottom: Interpolation between generated results.

2022; Shue et al., 2022). Therefore, we try to build a novel diffusion-based feed-forward framework for large-vocabulary 3D object generation with a single generative model. Specifically, in this paper, to handle the special challenges, we try to propose a **Diff**usion model with 3D-aware **T**rans**F**ormer, *i.e.*, **DiffTF** for general 3D object generation. In contrast to shape generation methods, our method can generate highly diverse 3D objects with semantic textures.

To meet the efficiency requirement of a) challenge, we adopt a revised sampling method in triplane fitting for boosting efficiency. Besides, to achieve better convergence, we constrain the distribution of triplanes in an appropriate range and make the values of triplanes smooth using normalization and strong regularization including total variation (TV) regularization (Dey et al., 2006). To handle the challenge caused by the large diversity and complex appearance of real-world objects, we regard all objects as a combination of generalized 3D prior knowledge and specialized 3D features. It is superior in two aspects: **1**) it can benefit from large-vocabulary objects compared with treating categories as independent properties. Since the generalized 3D knowledge is extracted from massive categories, it has strong generalized capability and robustness. **2**) it can decrease the complexity of generation. By separating the generalized information from different categories, the specialized feature extractor can pay more attention to the special information. To this end, we propose a novel 3D-aware transformer to achieve this goal. Since our transformer adopts a special share-weight cross-plane attention module, the extracted 3D prior knowledge is generalized to different planes of different categories. As for specialized 3D features, we apply different self-attention to integrate the 3D interdependencies into the encoded features. Consequently, our transformer can boost the performance in handling diverse objects. Additionally, to enhance 3D awareness and semantic information in triplanes for complicated categories, we built the 3D-aware encoder/decoder. Compared with the traditional 2D encoder that inevitably affects the 3D relations, our model can integrate 3D awareness into encoded features and enhance the 3D-related information. Finally, attributed to the effective generalized 3D knowledge and specialized 3D features extracted from various categories, DiffTF can generate impressive large-vocabulary 3D objects with rich and reasonable semantics.

To validate the effectiveness of our method in large-vocabulary 3D object generation, we conduct exhaustive experiments on two benchmarks, *i.e.*, ShapeNet (Chang et al., 2015) and OmniObject3D (Wu et al., 2023). Note that OmniObject3D contains 216 challenging categories with intricate geometry and texture, *e.g.*, pine cones, pitaya, various vegetables. As shown in Fig. 1, benefiting from the generalized 3D knowledge and the specialized 3D features extracted by 3D-aware modules, a single DiffTF model achieves superior performance in many complex and diverse categories, *e.g.*, litchi,

conch, pitaya, and dinosaur. Besides, our generated results have rich semantics and impressive performance in terms of topology and texture. The figure also intuitively proves that the latent space of our method is smooth semantically.

## 2 RELATED WORK

In this section, we introduce the recent related improvements in 3D generative models including GAN-based and diffusion-based methods, as well as transformer structure.

### 2.1 TRANSFORMER

In recent years, Transformer (Vaswani et al., 2017) has seen rapid progress in many fields including image recognition (Dosovitskiy et al., 2020b; Touvron et al., 2021), object detection (Carion et al., 2020; Zhu et al., 2020), tracking (Cao et al., 2021; 2022; 2023), segmentation (Zheng et al., 2021; Strudel et al., 2021), and image generation (Van den Oord et al., 2016; Jiang et al., 2021; Mo et al., 2023). Some works (Chen et al., 2020; Child et al., 2019) prove the remarkable of transformer when predicting the pixels autoregressively. Based on the masked token, MaskGIT. (Chang et al., 2022) achieve promising generation performance. DiT (Peebles & Xie, 2022) adopts the transformer as the backbone of diffusion models of images. Based on the 2D version, Mo et al. (2023) propose a DiT-3D for point cloud generation. Despite impressive progress in transformer-based generative models, they are optimized on a single or a few categories. In this paper, we propose a 3D-aware transformer for diverse real-world 3D object generation that can extract the generalized 3D prior knowledge and specialized 3D features from various categories.

### 2.2 3D GENERATION

With satisfying performance in 2D image synthesis, generative adversarial networks (GANs) inspire research in 3D via generating meshes (Gao et al., 2022), texture (Siddiqui et al., 2022), voxel (Chen et al., 2019), NeRFs (Chan et al., 2021; Gu et al., 2021; Niemeyer & Geiger, 2021; Schwarz et al., 2020; Zhou et al., 2021), Triplane (Chan et al., 2022), and point cloud (Achlioptas et al., 2018). Since the NeRFs and Triplane-based methods can adopt 2D images as supervision without any 3D assets, those two branches of methods have received much attention recently. By adding a standard NeRF model in the GAN model, Pi-GAN (Chan et al., 2021) and GRAF (Schwarz et al., 2020) can generate novel view synthesis of the generated 3D objects. Since the high memory loading and long training time of NeRF, adopting high-resolution images is hard to afford which impedes the generative performance. GIRAFFE (Niemeyer & Geiger, 2021) propose a downsample-upsample structure to handle this problem. By generating low-resolution feature maps and upsampling the feature, GIRAFFE (Niemeyer & Geiger, 2021) indeed improves the quality and resolution of output images. To address the 3D inconsistencies, StyleNeRF (Gu et al., 2021) design a convolutional stage to minimize the inconsistencies. To boost the training efficiency further, EG3D (Chan et al., 2022) propose an efficient 3D representation, *i.e.*, triplane. Due to its promising efficiency, in this work, we adopt the revised triplane as our 3D representation.

In contrast to GANs, diffusion models are relatively unexplored tools for 3D generation. A few valuable works based on NeRF (Poole et al., 2022; Li et al., 2022), point cloud (Nichol et al., 2022; Luo & Hu, 2021; Zeng et al., 2022), triplane (Shue et al., 2022; Wang et al., 2022; Gu et al., 2023) show the huge potential of diffusion model. DeamFusion (Poole et al., 2022) presents a method to gain NeRFs data and apply the generation based on the pre-train 2D text-image diffusion model. Similarly, NFD (Shue et al., 2022) views the triplane as the flattened 2D features and utilizes the 2D diffusion. It is indeed that adopting pre-train 2D diffusion can accelerate the training process. However, 2D prior knowledge also limits the capacity of the model in 3D-related information. To handle this problem, Rodin (Wang et al., 2022) proposes a 3D-aware convolution module in diffusion. Based on local CNN-based awareness, it can provide local 3D-related information for single-category generation. However, it is hard to maintain robustness when facing large categories for local 3D awareness. To this end, in this paper, we introduce global 3D awareness into our diffusion-based feed-forward model to extract the 3D interdependencies across the planes and enhance the relations within individual planes.

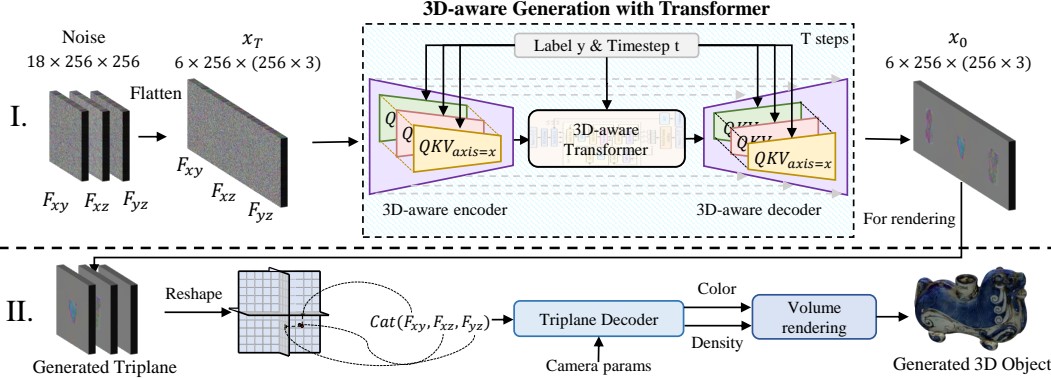

Figure 2: **Pipeline of DiffTF.** Sampling the 3D content from DiffTF has two individual steps: 1) using a trained diffusion model to denoise latent noise into triplane features, and 2) adopting the trained triplane decoder to decode the implicit features into the final 3D content.

## 3 METHODOLOGY

In this section, we will introduce the detailed structure of DiffTF. The sampling pipeline of DiffTF is shown in Fig. 2. More details about the two-step training process are released in the Appendix.

### 3.1 3D REPRESENTATION FITTING

An efficient and robust 3D representation is of significance to train a 3D diffusion model for large-vocabulary object generation. In this paper, we adopt the triplane representation inspired by (Chan et al., 2022). Specifically, it consists of three axis-aligned orthogonal feature planes, denoted as $F_{xy}, F_{yz}, F_{xz} \in \mathbb{R}^{C \times W \times H}$, and a multilayer perceptron (MLP) decoder for interpreting the sampled features from planes. By projecting the 3D coordinates on the triplane, we can query the corresponding features $F(p) = \text{Cat}(F_{xy}(p), F_{yz}(p), F_{xz}(p)) \in \mathbb{R}^{C \times 3}$ of any 3D position $p \in \mathbb{R}^{\times 3}$ via bilinear interpolation. given the position $p \in \mathbb{R}^3$ and view direction $d \in \mathbb{R}^2$, the view-dependent color $c \in \mathbb{R}^3$ and density $\sigma \in \mathbb{R}^1$ can be formulated as:

$$c(p, d), \sigma(p) = \text{MLP}_\theta(F(p), \gamma(p), d) , \tag{1}$$

where $\gamma(\cdot)$ represents the positional encoding. Detail structure of $\text{MLP}_\theta$ is shown in Appendix.

Since the fitted triplane is the input of diffusion, the triplanes of diverse real-world objects should be in the same space. Therefore, we need to train a robust category-independent shared decoder. More details about the decoder are released in Appendix. Besides, to constrain the values of triplanes, we adopt strong L2 regularization and TVloss regularization (Dey et al., 2006) as follows:

$$TV(F(p)) = \sum_{i,j} |F(p)[:, i+1, j] - F(p)[:, i, j]| + |F(p)[:, i, j+1] - F(p)[:, i, j]|. \tag{2}$$

Additionally, to further accelerate the speed of triplane fitting, the $p$ merely samples from the points in the foreground. To avoid the low performance in predicting background RGB, we set a threshold to control the ratio of only sampling from the foreground or sampling from the whole image. Besides, we introduce a foreground-background mask. According to the volume rendering(Max, 1995), the final rendered images $\hat{C}(p, d)$ and foreground mask $M(p)$ can be achieved by: $\hat{C}(p, d) = \sum_{i=1}^{N} T_i(1 - \exp(-\sigma_i \delta_i))c_i, where\ T_i = \exp(-\sum_{j=1}^{i-1} \sigma_j \delta_j), \hat{M}(p) = \sum_{i=1}^{N} T_i(1 - \exp(-\sigma_i \delta_i))$ Therefore, the training objective of triplane fitting can be formulated as:

$$\mathcal{L} = \sum_{i}^{M} (\text{MSE}(\hat{C}(p, d), G_c) + \text{MSE}(\hat{M}(p, d), G_\sigma)) + \lambda_1(\text{TV}(F(p))) + \lambda_2(||F(p)||_2^2) , \tag{3}$$

where $G_c$ and $G_\sigma$ represent the ground-truth of RGB and alpha while $\lambda_1$ and $\lambda_2$ are coefficients.

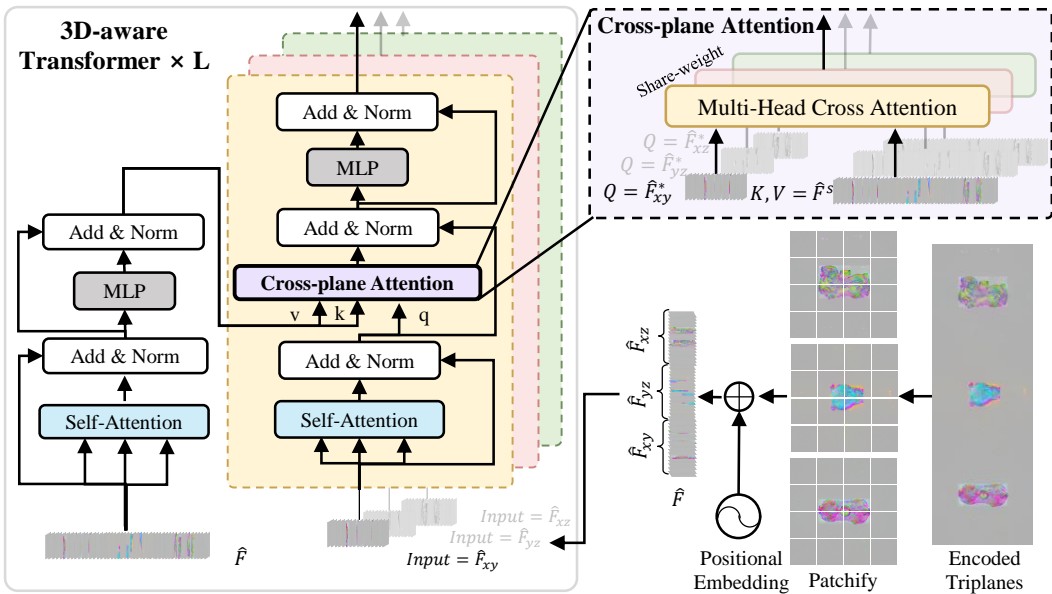

Figure 3: The detailed structure of our proposed 3D-aware modules. We take the feature from the xy plane $\hat{F}_{xy}$ as an example. Relying on the extracted generalizable 3D knowledge and specialized one, our model can achieve impressive adaptivity among various categories.

## 3.2 3D-AWARE DIFFUSION WITH TRANSFORMER

### 3.2.1 3D-AWARE ENCODER/DECODER

In general, low-resolution features tend to contain richer semantic information and exhibit greater robustness, while high-resolution features are beneficial for capturing detailed features. Therefore, to enhance the 3D-related information in triplanes and avoid compute-intensive operation in the 3D-aware transformer, we propose a new 3D-aware encoder/decoder to encode the original triplanes $F^t = \mathrm{Cat}(F^t_{xy}, F^t_{yz}, F^t_{xz}) \in \mathbb{R}^{C \times W \times 3H}$ with 3D awareness.

To avoid compute-intensive operations in the encoder/decoder, we adopt a single convolution to decrease the resolution of features. Note that since the triplanes have 3D relations, the convolution should be performed on individual planes respectively. For simplification, we denote the output of convolution as $\overline{F}^t \in \mathbb{R}^{C \times W' \times 3H'}, W' = W/n, H' = H/n$, where $n$ represents the stride of convolution. Following the patchify module in (Dosovitskiy et al., 2020a), we adopt a patchify layer (denote as $\mathcal{G}$) to convert the spatial features into a sequence of $M$ tokens. Thus, given the patch size $ps$, the dimension of output feature sequences can be formulated as: $\overline{F}' = \mathcal{G}(\overline{F}^t) \in \mathbb{R}^{C \times 3M}$, where $M = W'/ps * H'/ps$. Since the attention operation is position-independent, the learnable positional embedding is applied after the patchify module (denoted the output as $\overline{F}$). Considering triplanes from different categories may have differences in feature space, we introduce the conditional information using an adaptive norm layer inspired by (Peebles & Xie, 2022).

To extract the generalized 3D knowledge across different planes, we introduce the cross-plane attention whose details are shown in Fig. 3. It takes the target plane (for example xy-plane) as the query while treating the whole triplane as key and values. The 3D constraint in feature space between xy-plane and three planes can be obtained as:

$$\widetilde{F}_{xy} = \overline{F}_{xy} + \alpha_1 * \mathrm{MultiHead}(\overline{F}_{xy}, \overline{F}, \overline{F}) \quad , \tag{4}$$

where $\alpha_1$ represents the calibration vector obtained by a MLP. To maintain the 3D-related information, we adopt a share-weight attention module. Consequently, the encoded feature with 3D awareness can be concatenated: $\widetilde{F} = \mathrm{Cat}(\widetilde{F}_{xy}, \widetilde{F}_{yz}, \widetilde{F}_{xz})$, where $\widetilde{F}_{yz}$ and $\widetilde{F}_{xz}$ are the 3D constraint features from yz and xz planes respectively.

In the end, the sequence features can be restored to spatial features via an integration layer (denoted as $\mathcal{G}^{-1}$): $\widetilde{F}^t = \mathcal{G}^{-1}(\widetilde{F}) \in \mathbb{R}^{C \times W' \times H'}$. The detail structure is released in Appendix.

### 3.2.2 3D-AWARE TRANSFORMER

Benefiting from the 3D-aware transformer encoder, the 3D-related information is enhanced in the encoded high-level features. To extract the global generalized 3D knowledge and aggregate it with specialized features further, we build the 3D-aware transformer illustrated in Fig. 3. It consists of self- and cross-plane attention. The former aims to enhance the specialized information of individual planes while the latter concentrates on building generalized 3D prior knowledge across different planes.

Similar to the 3D-aware encoder, we adopt the patchify module and the learnable positional embedding. For clarification, we denote the input of the 3D-aware transformer as $\hat{F}$. Since the encoded features have in terms of plane-dependent and plane-independent information, we build the transformer in two branches. Specifically, the self-attention module in the left branch can build interdependencies within individual planes. By exploiting 2D high-level features in different planes, our transformer will exploit the rich semantic information. Therefore, the output of enhanced features can be formulated as follows:

$$\hat{F}^s = \text{Norm}(\hat{F}_1 + \text{MLP}(\hat{F}_1)), \quad \hat{F}_1 = \text{Norm}(\hat{F} + \text{MultiHead}(\hat{F}, \hat{F}, \hat{F})), \tag{5}$$

where $\text{Norm}$ represents the layer normalization.

As for the right branch, it focuses on extracting the global 3D relations across different planes. Similarly, to enhance the explicit 2D features in each plane, we adopt an additional self-attention module before cross-plane attention. Meanwhile, to avoid the negative influence of 2D semantic features, we apply the residual connection. Consequently, the 3D-related information of xy planes can be formulated as:

$$\begin{aligned}
\hat{F}_{xy}^{final} &= \text{Norm}(\hat{F}_{xy}^c + \text{MLP}(\hat{F}_{xy}^c) \\
\hat{F}_{xy}^c &= \text{Norm}(\hat{F}_{xy}^* + \text{MultiHead}(\hat{F}_{xy}^*, \hat{F}^s, \hat{F}^s)) \\
\hat{F}_{xy}^* &= \text{Norm}(\hat{F}_{xy} + \text{MultiHead}(\hat{F}_{xy}, \hat{F}_{xy}, \hat{F}_{xy}))
\end{aligned} \tag{6}$$

By concatenating the features from three planes and integration layer ($\mathcal{G}^{-1}$) similar to 3D-aware encoder, the final features containing 3D-related information can be restored to the spatial features: $\hat{F}_d = \mathcal{G}^{-1}(\text{Cat}(\hat{F}_{xy}^{final}, \hat{F}_{yz}^{final}, \hat{F}_{xz}^{final})) \in \mathbb{R}^{C \times W' \times H'}$.

## 4 EXPERIMENTS

### 4.1 IMPLEMENTATION DETAILS

**Datasets.** Following most previous works, we use the ShapeNet (Chang et al., 2015) including Chair, Airplane, and Car for evaluating the 3D generation which contains 6770, 4045, and 3514 objects, respectively. Additionally, to evaluate the large-vocabulary 3D object generation, we conduct the experiments on a most recent 3D dataset, OmniObject3D (Wu et al., 2023). OmniObject3D is a large-vocabulary real scanned 3D dataset, containing 216 challenging categories of 3D objects with high quality and complicated geometry, *e.g.*, toy, fruit, vegetable, and art sculpture.

**Evaluation Metrics.** Following prior work (Müller et al., 2022; Shue et al., 2022), we adopt two well-known 2D metrics and two 3D metrics: a) Fréchet Inception Distance (Heusel et al., 2017) (FID-50k) and Kernel Inception Distance (Bińkowski et al., 2018) (KID-50k); b) Coverage Score (COV) and Minimum Matching Distance (MMD) using Chamfer Distance (CD). All metrics are evaluated at a resolution of $128 \times 128$. More details are released in Appendix.

### 4.2 COMPARISON AGAINST STATE-OF-THE-ART METHODS

In this section, we compare our methods with state-of-the-art methods, including two GAN-based methods: EG3D (Chan et al., 2022), GET3D (Gao et al., 2022) and two diffusion-based methods: DiffRF (Müller et al., 2022), and NFD (Shue et al., 2022).

**Large-vocabulary 3D object generation on OmniObject3D.** The class-conditional quantitative results on OmniObject3D are shown in Table 1. Compared with NFD w/texture that uses the 2D CNN

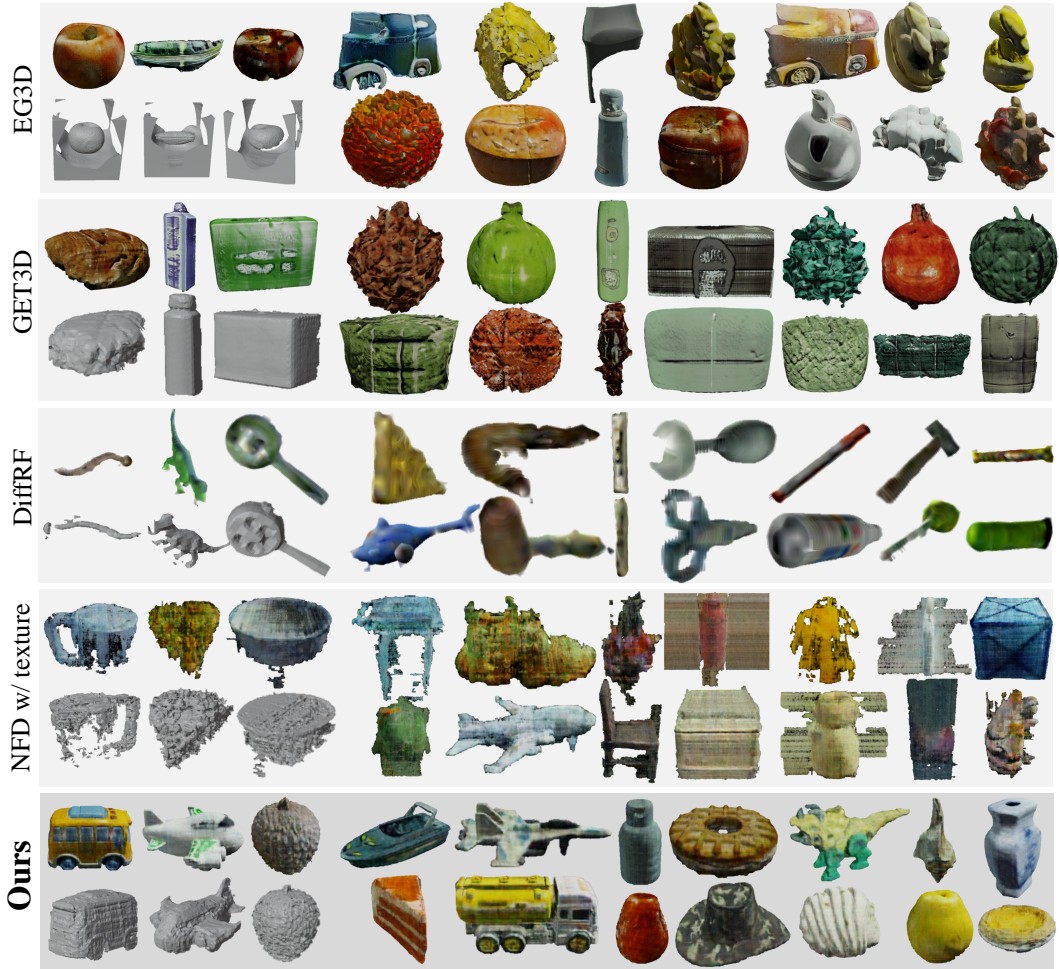

Figure 4: **Qualitative comparisons to the SOTA methods in terms of generated 2D images and 3D shapes on OmniObject3D.** Compared with other SOTA methods, our generated results are more realistic with richer semantics.

| Methods | FID↓ | KID(%)↓ | COV(%)↑ | MMD(‰)↓ |
|---|---|---|---|---|
| EG3D (Chan et al., 2022) | 41.56 | 1.0 | 14.14 | 28.21 |
| GET3D (Gao et al., 2022) | 49.41 | 1.5 | 24.42 | 13.57 |
| DiffRF (Müller et al., 2022) | 147.59 | 8.8 | 17.86 | 16.07 |
| NFD w/ texture (Shue et al., 2022) | 122.40 | 4.2 | 33.15 | 10.92 |
| **DiffTF (Ours)** | **25.36** | **0.8** | **43.57** | **6.64** |

Table 1: **Quantitative comparison of conditional generation on the OmniObject3D dataset.** DiffTF outperforms other SOTA methods in terms of 2D and 3D metrics by a large margin.

diffusion, our 3D-aware transformer achieves promising improvement in both texture and geometry metrics, especially in texture metric. It proves the effectiveness of the global 3D awareness introduced by our transformer. Compared to DiffRF adopting 3D CNN on voxel, the diffusion on triplane features can benefit from higher-resolution representation to boost performance. Additionally, our methods outperform the SOTA GAN-based methods with a significant improvement. The visualization illustrated in Fig 4 intuitively demonstrates the superiority of DiffTF in terms of geometry and texture. Particularly, our generated objects have rich semantics that make our results easy to determine the category. More qualitative results are released in Appendix.

**Single-category generation on ShapeNet.** To validate our methods in a single category, we provide the quantitative and qualitative results on ShapeNet (Chang et al., 2015) in Fig 5 and Table 2. Compared with other recent SOTA methods, our methods achieve significant improvements in terms of all metrics in three categories. Our method is more stable and robust in all three categories. As

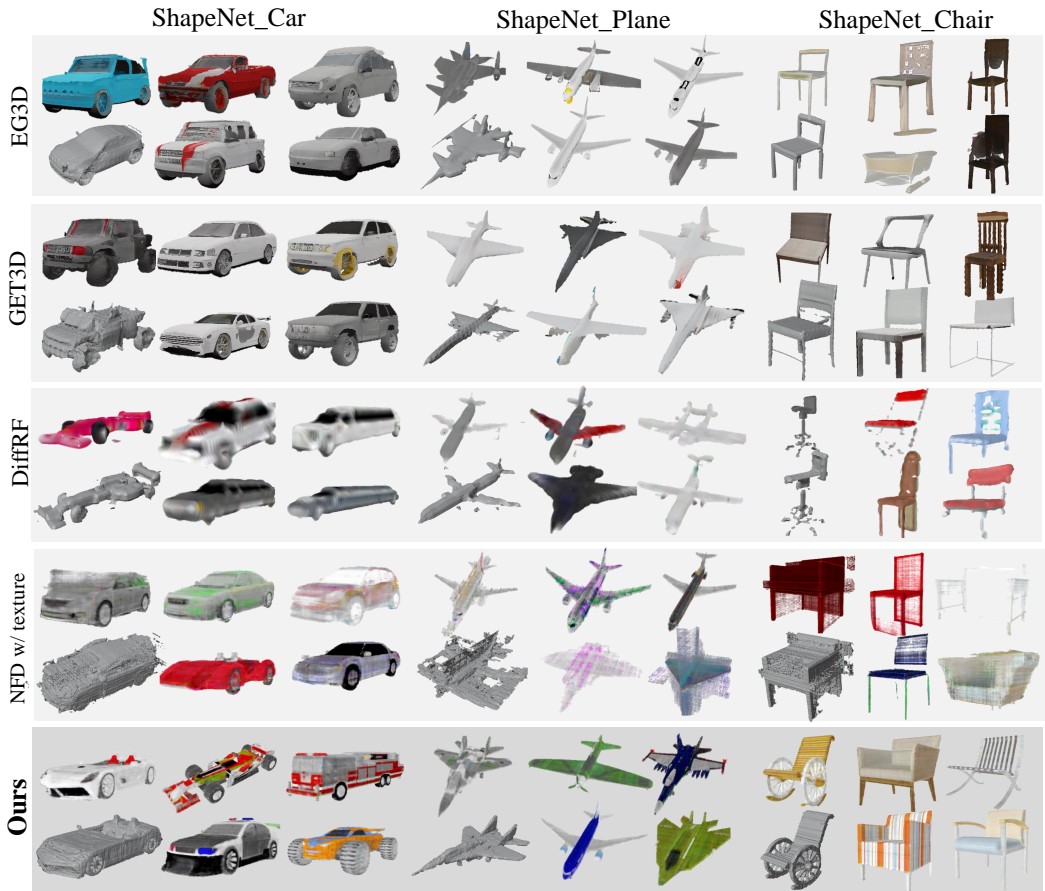

Figure 5: **Qualitative comparison of DiffTF against other SOTA methods on ShapeNet**. It intuitively illustrates the promising performance of our method in texture and topology.

| Category | Method | FID↓ | KID(%)↓ | COV(%)↑ | MMD(‰)↓ |
|---|---|---|---|---|---|
| Car | EG3D (Chan et al., 2022) | 45.26 | 2.2 | 35.32 | 3.95 |
| | GET3D (Gao et al., 2022) | 41.41 | 1.8 | 37.78 | 3.87 |
| | DiffRF (Müller et al., 2022) | 75.09 | 5.1 | 29.01 | 4.52 |
| | NFD w/ texture (Shue et al., 2022) | 106.06 | 5.5 | 39.21 | 3.85 |
| | **DiffTF (Ours)** | **36.68** | **1.6** | **53.25** | **2.57** |
| Plane | EG3D (Chan et al., 2022) | 29.28 | 1.6 | 18.12 | 4.50 |
| | GET3D (Gao et al., 2022) | 26.80 | 1.7 | 21.30 | 4.06 |
| | DiffRF (Müller et al., 2022) | 101.79 | 6.5 | 37.57 | 3.99 |
| | NFD w/ texture (Shue et al., 2022) | 126.61 | 6.3 | 34.06 | 2.92 |
| | **DiffTF (Ours)** | **14.46** | **0.8** | **45.68** | **2.58** |
| Chair | EG3D (Chan et al., 2022) | 37.60 | 2.0 | 19.17 | 10.31 |
| | GET3D (Gao et al., 2022) | 35.33 | 1.5 | 28.07 | 9.10 |
| | DiffRF (Müller et al., 2022) | 99.37 | 4.9 | 17.05 | 14.97 |
| | NFD w/ texture (Shue et al., 2022) | 87.35 | 2.9 | 31.98 | 7.12 |
| | **DiffTF (Ours)** | **35.16** | **1.1** | **39.42** | **5.97** |

Table 2: **Quantitative comparison of unconditional generation on the ShapeNet**. It proves the impressive performance of our DiffTF on the single-category generation.

shown in Fig 5, our generated results contain more detailed semantic information which makes our generated results more realistic. More qualitative results are released in Appendix.

### 4.3 ABLATION STUDY

In this section, we conduct the ablation studies mainly in two ways: 1) diffusion: w/ and w/o our proposed 3D-aware modules; 2) triplane fitting: w/ and w/o normalization and regularization.

| TP Regu | TP Norm | CP | Ori-TF | CP-TF | FID↓ | KID(%)↓ | COV(%)↑ | MMD(‰)↓ |
|---------|---------|-----|--------|-------|-------|---------|---------|---------|
| ✗ | ✗ | ✗ | ✗ | ✗ | 125.21 | 2.6 | 21.31 | 16.10 |
| ✓ | ✗ | ✗ | ✗ | ✗ | 113.01 | 2.3 | 30.61 | 12.04 |
| ✓ | ✓ | ✗ | ✗ | ✗ | 109.81 | 2.0 | 33.15 | 10.92 |
| ✓ | ✓ | ✓ | ✗ | ✗ | 39.13 | 1.4 | 37.15 | 9.93 |
| ✓ | ✓ | ✓ | ✓ | N/A | 52.35 | 2.0 | 36.93 | 9.97 |
| ✓ | ✓ | ✓ | N/A | ✓ | **25.36** | **0.8** | **43.57** | **6.64** |

Table 3: **Ablation studies on OmniObject3D**. The TP Regu and TP Norm represent the triplane regularization and triplane normalization. Besides, we denote the proposed cross-plane attention in the encoder, original transformer and our 3D-aware transformer as CP, Ori-TF, and CP-TF.

**Ablating normalization and regularization.** Since the triplane fitting is the foundation of the diffusion model, we also study the influence of normalization and regularization in triplane fitting and diffusion training. As shown in Fig. 6, the triplane features are more smooth and clear. With effective regularization, the generated objects have better shapes and geometry information. Since the distribution of triplanes is overwide (from -10 to 10) without constraint, it is essential to adopt normalization to accelerate the convergence. As illustrated in Table 3, by adopting the preprocess on triplane features, the diffusion model can get better performance.

**Studies of 3D-aware transformer modules.** Compared with 2D CNN, our 3D-aware encoder/decoder can encode the triplanes while enhance the 3D relations. As shown in Fig. 6, attributing to the enhanced 3D features, it can raise the overall generative performance. Notably, since the original transformer merely utilizes 2D self-attention, the introduction of the 2D interdependencies will break the latent 3D relations built by the encoder, thereby impeding the performance shown in Table 3. In contrast to the original transformer, our 3D-aware transformer can effectively extract generalized 3D-related information across planes and aggregate it with specialized one for large-vocabulary generation. In conclusion, attributed to extracted generalized 3D knowledge and specialized 3D features, our 3D-aware transformer is capable of strong adaptivity for large-vocabulary 3D objects. More qualitative and network architecture comparisons are released in Appendix.

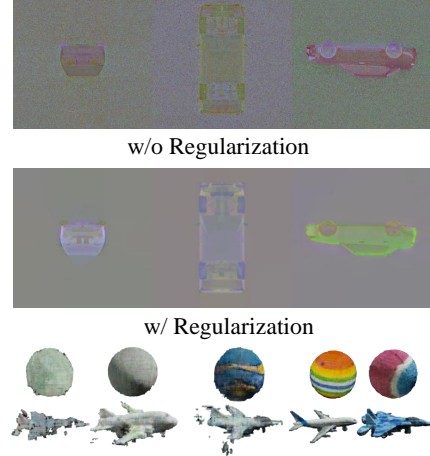

w/o Regularization

w/ Regularization

Baseline    Baseline+CP    Baseline+CP+Ori-TF    **Baseline+CP+CP-TF**

Figure 6: Top: Ablations on regularization. Bottom: Ablations on 3D-aware modules.

## 5 CONCLUSION

In this paper, in contrast to the prior work optimizing the generative model on a single category, we propose a diffusion-based feed-forward framework for synthesizing various categories of real-world 3D objects with a single model. To handle the special challenges in large-vocabulary 3D object generation, we 1) improve the efficiency of triplane fitting; 2) introduce the 3D-aware transformer to boost the generative performance in diverse categories; 3) propose the 3D-aware encoder/decoder to handle the categories with complicated topology and texture. Separating generalized and specialized information from 3D objects can 1) enable more effective utilization of diverse 3D objects; 2) assist the specialized feature extractor focus on specialized features, thereby reducing the complexity of the large-vocabulary generation. Based on the extracted generalized and specialized 3D prior knowledge, our DiffTF can generate diverse high-quality 3D objects with rich semantics. The exhaustive comparisons against SOTA methods validate the promising performance of our methods. We believe that our work can provide valuable insight for the 3D generation community.

**Limitation.** While our method makes a significant improvement in 3D generation, it still has some limitations: 1) the triplane fitting speed (∼3 minutes per object) is still time-consuming when scaled up to 1 million objects; 2) the proper noise scheduling scheme for triplane generation is not well-analyzed, impeding the improvement of generative performance, especially in detail. In our future work, we will try to handle those problems to boost the generative performance further.

**Ethics statement.** In this paper, we proposed a novel large-vocabulary 3D diffusion model that generates high-quality textures and geometry with a single model. Therefore, our method could be extended to generate DeepFakes or generate fake images or videos of any person intending to spread misinformation or tarnish their reputation.

ACKNOWLEDGMENTS

This study is supported by Shanghai Artificial Intelligence Laboratory (P23KS00010, 2022ZD0160201), the Ministry of Education, Singapore, under its MOE AcRF Tier 2 (MOE-T2EP20221- 0012), NTU NAP, and under the RIE2020 Industry Alignment Fund – Industry Collaboration Projects (IAF-ICP) Funding Initiative, as well as cash and in-kind contribution from the industry partner(s).

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
