# SUPPLEMENTARY MATERIAL FOR LARGE-VOCABULARY 3D DIFFUSION MODEL WITH TRANSFORMER

## A APPENDIX

In this supplementary material, we report the implementation details about 1) training, *i.e.*, triplane fitting and diffusion training; 2) related setting

### A.1 IMPLEMENTATION DETAILS

#### A.1.1 TRAINING DETAILS

**Diffusion.** We adopt the cross-plane attention layer in the 3D-aware encoder when the feature resolution is 64, 32, and 16. We adopt 8, 4, and 2 as the patch size in the encoder/decoder. The patch size and number of the 3D-aware transformer layers are set to 2 and 4, respectively. Following the prior work (Müller et al., 2022; Shue et al., 2022), we adopt T=1000 during training and T=250 for inference. Our diffusion model is trained using an Adam optimizer with a learning rate of 1e-4 which will decrease from 1e-4 to 1e-5 in linear space. We apply a linear beta scheduling from 0.0001 to 0.01 at 1000 timesteps. Besides, we adopt the $\epsilon$ as the objective of our diffusion model. We train our model for about 3 days on 32 NVIDIA A100 GPUs.

**Triplane fitting.** Our implementation is based on the PyTorch framework. The dimension of the triplane is $18 \times 256 \times 256$. Note that $\lambda_1$ and $\lambda_2$ are set to 1e-4 and 5e-5 for training the share-weight decoder. We train our shared decoder using 8 GPUs for 24 hours. After getting the decoder, $\lambda_1$ and $\lambda_2$ are set to 0.5 and 0.1 for triplane fitting. To improve the robustness of the shared decoder, we adopt the one-tenth learning rate (1e-2) during the training while the learning rate of the triplane feature is set to 1e-1.

#### A.1.2 DETAILED WORKFLOW OF DIFFTF

**Training:** The training process consists of two steps illustrated in Fig. 1. In the **step I**, *i.e.*, triplane fitting, the objective is to obtain the diverse triplane features and robust triplane decoder. Therefore, for clarification, we divide **step I** into two subtasks: **step I-I** training shared decoder and **step I-II** optimizing triplanes from diverse 3D objects. To maintain the robustness of the decoder, we adopt around 20 percent diverse and high-quality objects for optimizing the shared decoder in the **step I-I**. Then, in **step I-II**, we adopt the trained decoder with frozen parameters to merely fit the triplanes. After obtaining the fitted triplanes, we can use them as the ground truth to train the 3D-aware transformer-based diffusion model.

**Sampling:** Similar to the training process, sampling the 3D content from DiffTF has two individual steps: 1) using a trained diffusion model to denoise latent noise into triplane features, and 2) adopting the trained triplane decoder to decode the implicit features into the final 3D content.

### A.2 DATA

**Training data** To train the triplane and shared decoder on ShapeNet, we use the blender to render the multi-view images from 195 viewpoints. Those points sample from the surface of a ball with a 1.2 radius. Similarly, we use the blender to render the 5900+ objects from 100 different viewpoints to fit the triplane and decoder on OmniObject3D following (Wu et al., 2023).

**Evaluation** The 2D metrics are calculated between 50k generated images and all available real images. Furthermore, For comparison of the geometrical quality, we sample 2048 points from the

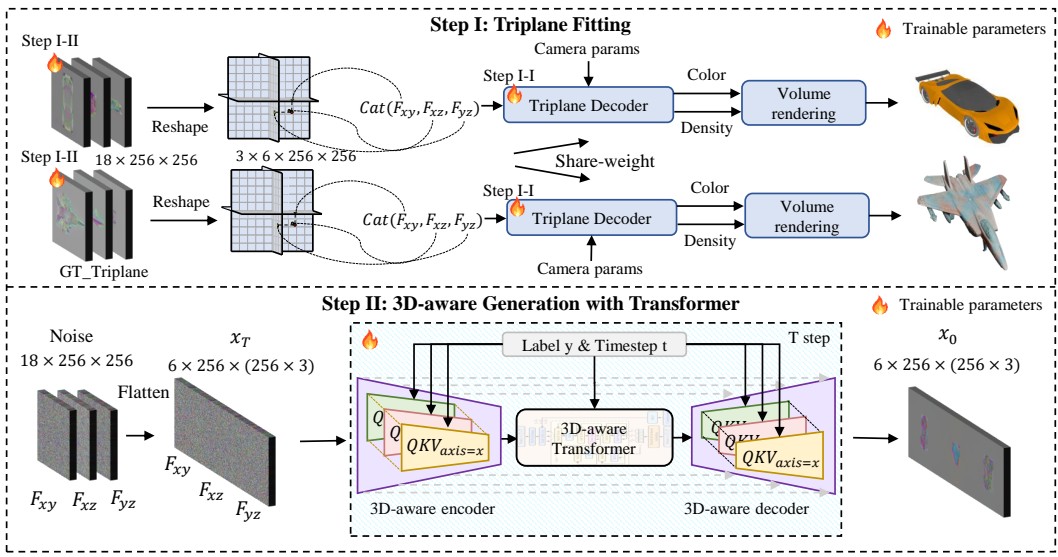

Figure 1: **An overview of our training process.** It is composed of two steps: **Step I**) training the share-weight decoder (Step I-I) and fitting the triplane features (Step I-II), and **Step II**) optimizing our 3D-aware transformer diffusion using the trained triplanes in Step I-II.

surface of 5000 objects and apply the Coverage Score (COV) and Minimum Matching Distance (MMD) using Chamfer Distance (CD) as follows:

$$CD(X, Y) = \sum_{x \in X} \min_{y \in Y} ||x - y||_2^2 + \sum_{y \in Y} \min_{x \in X} ||x - y||_2^2,$$

$$COV(S_g, S_r) = \frac{|\{\arg \min_{Y \in S_r} CD(X, Y) | X \in S_g\}|}{|S_r|},$$

$$MMD(S_g, S_r) = \frac{1}{|S_r|} \sum_{Y \in S_r} \min_{X \in S_g} CD(X, Y) \qquad (1)$$

where $X \in S_g$ and $Y \in S_r$ represent the generated shape and reference shape.

Note that we use 5k generated objects $S_g$ and all available real shapes $S_r$ to calculate COV and MMD. For fairness, we normalize all point clouds by centering in the original and recalling the extent to [-1,1]. Coverage Score aims to evaluate the diversity of the generated samples, MMD is used for measuring the quality of the generated samples. 2D metrics are evaluated at a resolution of $128 \times 128$. For the Car in ShapeNet, since the GT data contains intern structures, we thus only sample the points from the outer surface of the object for results of all methods and ground truth.

**Details about SOTA methods** Since the official NFD merely generates the 3D shape without texture, we reproduce the NFD w/ texture as our baseline. Besides, we use the official code and the same rendering images to train the EG3D and GET3D while the DiffRF and NFD adopt our reproduced code. Note that because of adopting pose condition, official EG3D doesn't support class-conditional generation. Therefore, to maintain the fairness of our evaluations, other class-conditional generative models including DiffTF adopt a random class-conditional input.

**Details about Interpolation** Song et al. (2020) proves smooth interpolation in the latent space of diffusion models can be achieved by interpolation between noise tensors before they are iteratively denoised by the model. Therefore, we sample from our model using the DDIM method. To guarantee the same distribution of the interpolation samples, we adopt spherical interpolation.

### A.3    ADDITIONAL DETAILS IN METHODOLOGY

The detailed structure of our shared decoder in triplane fitting is shown in Fig 2 while Figure 3 illustrates the structure of our 3D-aware modules.

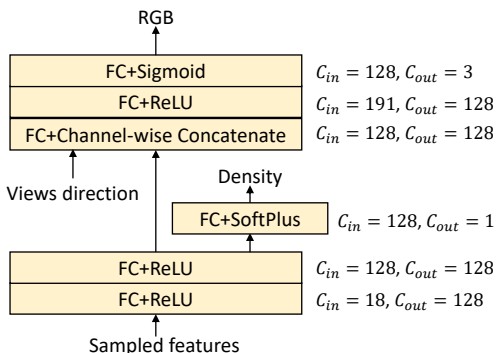

Figure 2: The MLP structure of shared decoder.

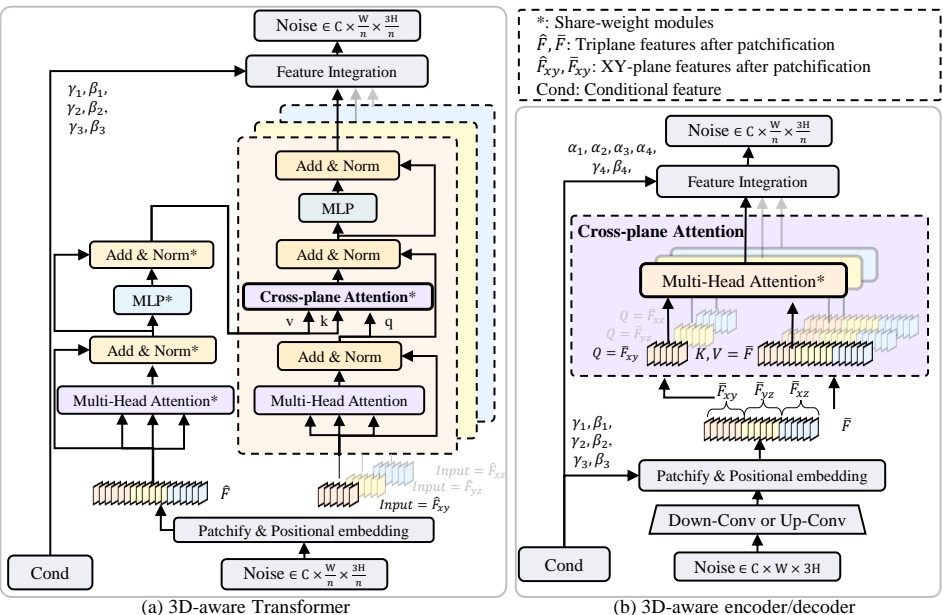

(a) 3D-aware Transformer                    (b) 3D-aware encoder/decoder

Figure 3: **The detailed structure of our proposed 3D-aware modules**. We take the feature from the xy plane as an example. The left module (a) aims to extract the global 3D prior knowledge across planes. The right one (b) tries to efficiently encode the triplanes while maintaining the 3D-related information via a single cross-plane attention module.

### A.3.1 REVISIT MULTI-HEAD ATTENTION AND DDPMS ON 3D GENERATION

**Multi-Head Attention:** Multi-Head Attention is a fundamental component in transformer structure (Vaswani et al., 2017) which can be formulated by:

$$\begin{aligned}
\text{MultiHead}(Q, K, V) &= \Big(\text{Cat}(H_{att}^1, ..., H_{att}^N)\Big) W \\
H_{att}^n &= \text{Attention}(QW_q^n, KW_k^n, VW_v^n) \\
\text{Attention}(Q, K, V) &= \text{Softmax}(\frac{QK^{\mathrm{T}}}{\sqrt{d}})V
\end{aligned} \quad , \tag{2}$$

where $Q, K,$ and $V$ represent the query, key, and value in the attention operation, $W \in \mathbb{R}^{C_i \times C_i}$, $W_q^n \in \mathbb{R}^{C_i \times C_h}$, $W_k^n \in \mathbb{R}^{C_i \times C_h}$, and $W_v^n \in \mathbb{R}^{C_i \times C_h}$ are learnable weights, and $\sqrt{d}$ is the scaling factor to avoid gradient vanishing.

**DDPM:** To solve the generation problem, denoising diffusion probabilistic models (DDPMs) define the forward and reverse process which transfers the generation problem into predicting noise.

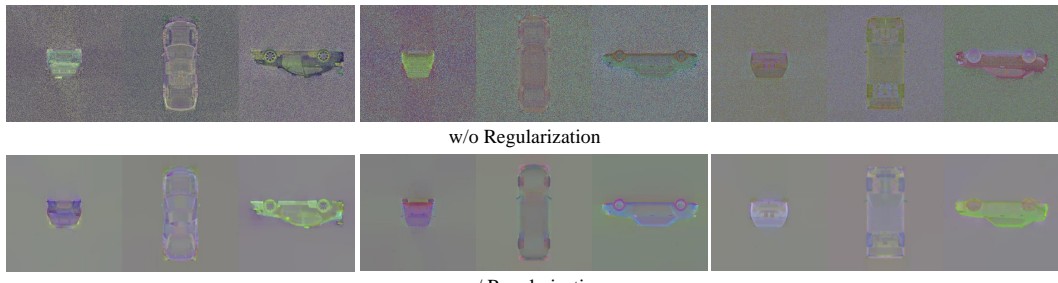

w/o Regularization

w/ Regularization

Figure 4: Visualization of triplane. Top: triplane fitting without TVloss and L2 regularization. Bottom: triplane fitting with TVloss and L2 regularization. It illustrates that by effective regularization, the triplane features are smooth and clear which is helpful for the next training.

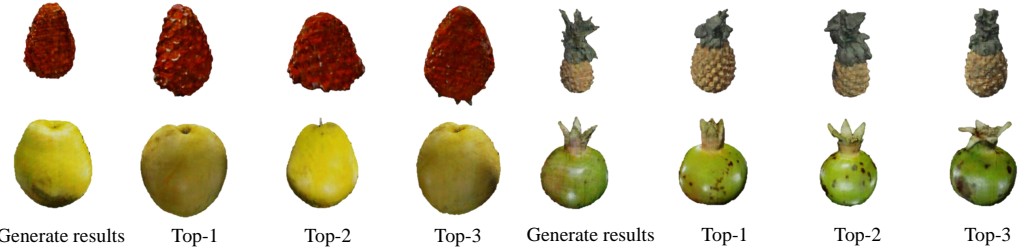

Generate results    Top-1    Top-2    Top-3    Generate results    Top-1    Top-2    Top-3

Figure 5: Nearest Neighbor Check on OmniObject3D. We compare our generated results and the most similar top 3 objects from the training set.

The forward process represents the process of applying noise to real data $x_0$ as $q(x_t|x_{t-1}) = \mathcal{N}(x_t; \sqrt{1-\beta_t}x_{t-1}, \beta_t \mathbf{I})$, where $\beta_t$ and $\mathbf{I}$ represents the forward process variances and unit matrix. For clarification, we assume there are $T$ steps in the forward process. Thus, the features with different noised levels can be denoted as $x_T, x_{T-1}, ..., x_0$, where $x_T$ is sampled from a standard Gaussian noise. Based on the relationship between two continuous steps, we have $x_t(x_0, \epsilon) = \sqrt{\overline{\alpha}_t}x_0 + \sqrt{1-\overline{\alpha}_t}\epsilon$, where $\epsilon \sim \mathcal{N}(0, \mathbf{I}), \alpha_t = 1 - \beta_t$ and $\overline{\alpha}_t = \prod_{i=1}^{t} \alpha_i$

During the reverse process, the diffusion model aims to predict the $p_\theta(x_{t-1}|x_t)$, $\theta$ represents the learnable parameters of the diffusion network. Based on Bayes' theorem and specific parameterization, the $p_\theta(x_{t-1}|x_t)$ can be formulated as: $p_\theta(x_{t-1}|x_t) = \mathcal{N}(x_{t-1}; \mu_\theta(x_t, t), \Sigma_\theta(x_t, t))$, where $\mu_\theta(x_t, t) = \frac{1}{\sqrt{\alpha_t}}(x_t - \frac{\beta_t}{\sqrt{1-\overline{\alpha}_t}}\epsilon_\theta(x_t, t))$ and $\Sigma_\theta(x_t, t) = \sigma_t^2 \mathbf{I}$, where $\sigma_t^2 = \frac{1-\overline{\alpha}_{t-1}}{1-\overline{\alpha}_t}\beta_t$. In the end, the objective of the reverse process is transferred to predict $\epsilon$. Thus the objective of the training is to minimize the loss function as $\mathcal{L}_{diff} = \mathbb{E}_{t,x_0,\epsilon}\left[||\epsilon - \epsilon_\theta(x_t, t)||^2\right]$.

Note that the symbols mentioned in Sec. A.3.1 hold distinct meanings compared to the ones in the main paper.

## A.4   ADDITIONAL RESULTS

**Nearest Neighbor Check using CLIP** To validate the generative capability of our method, we perform the nearest neighbor check on OmniObject3D. As shown in Fig. 5, our method can generate some novel objects. We achieve the nearest neighbor check via the CLIP model. After obtaining the CLIP features, we chose the top 3 results by measuring cosine distances.

**More qualitative results** Additional qualitative results are shown in Fig 8, Fig 9, and Fig. 10. Our generated results contain more detailed semantic information which makes our generated results more realistic. Furthermore, the performance of other methods on class-conditional generation is shown in Fig. 11. It clearly i

**Ablations about triplane fitting** As shown in Fig. 4, we provide additional comparison between w/ and w/o strong regularization. Besides, we report the distribution of triplane before the normalization shown in Fig. 13. It illustrates that the values of triplanes are from -10 to 10 if we adopt no

Network Architecture

Figure 6: Ablations on sampling strategy.

Figure 7: Different architectures in ablation studies.

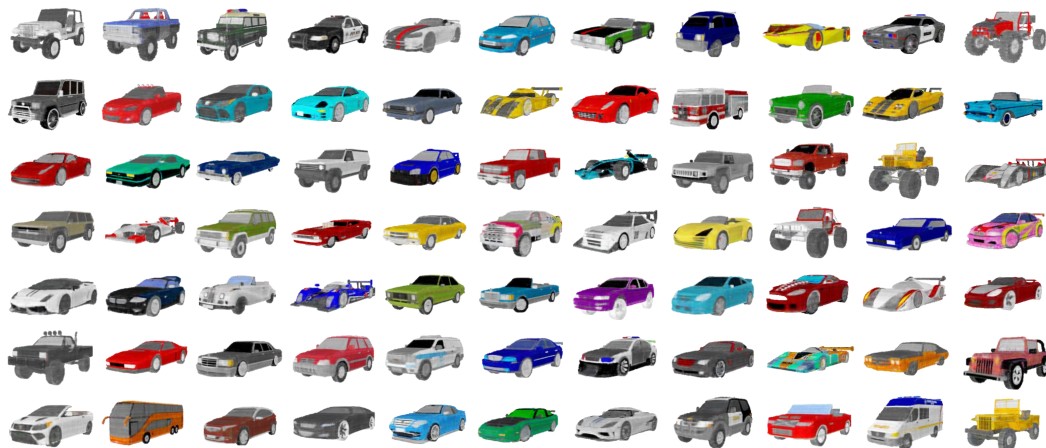

Figure 8: Qualitative results on ShapeNet_Car.

normalization, blocking the optimization of the diffusion model. Therefore, it is essential to scale to the range of -1 to 1 to achieve a better convergence of the diffusion model. In addition, we study the effectiveness of the new sampling strategy in triplane fitting. Notably, the PSNR in Fig. 6 is measured merely in the foreground area. With the new sampling strategy, the speed of our triplane fitting is raised further.

**Studies of 3D-aware transformer modules.** To intuitively demonstrate the superiority of our proposed modules, we release the performance comparison between different structures illustrated in Fig 12. It strongly verifies the effectiveness of our modules. To clarify, the comparison of different network architectures is shown in Fig 7. Furthermore, we also compare our methods with other 3D-aware modules, *i.e.*, 3D-aware convolution Wang et al. (2022). Experiment results in Table. 1 demonstrate the impressive generative performance of our novel 3D-aware modules compared with 3D-aware convolution in large-vocabulary 3D geenration.

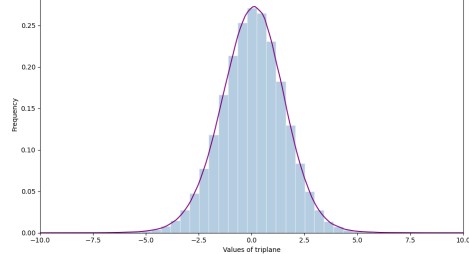

Figure 13: Distribution of triplane values before normalization.

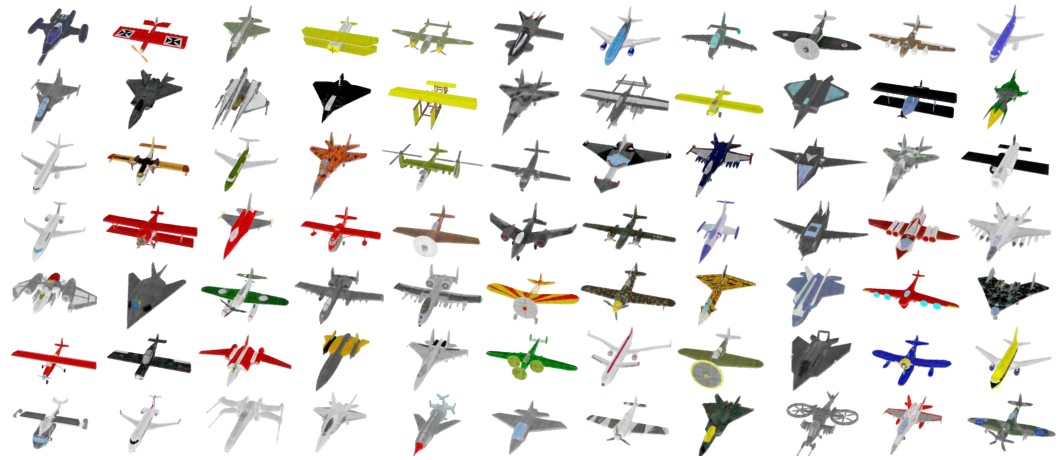

Figure 9: Qualitative results on ShapeNet_Plane.

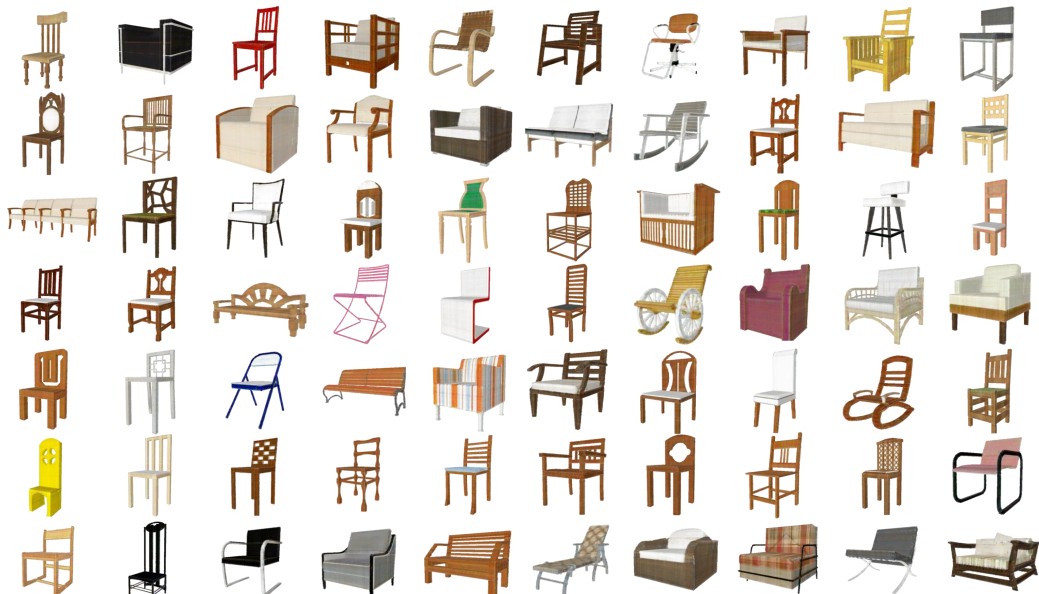

Figure 10: Qualitative results on ShapeNet_Chair.

**Comparison of single-category and multi-category models** We also compare the performance of different setting models. As shown in Fig. 4, our model trained on multi-category objects (class-conditional) can maintain an impressive generative performance.

**User studies** In addition to the 2D and 3D metrics mentioned above, we also perform a user study and report human's preference on rendered images. We analyze the generated results from three aspects, *i.e.*, overall performance, texture, and geometry. The results in Table 2 prove the superior performance of our method. Our method gains a significant improvement in three aspects against the SOTA methods.

**Comparison of parameters and FLOPs** Additionally, we compare DiffTF against other diffusion-based methods in parameters and FLOPs. Since the NFD merely involves the 2D CNN, the architecture is more efficient than DiffRF (using 3D CNN on mesh) and our DiffTF (using 3D-aware modules). Without the most efficient structure, our DiffTF is capable of generating a 3D object within 30 seconds. Additionally, to accelerate the speed of sampling further, we can adopt the DDIM with 50 diffusion steps. Adopting this setting, DiffTF can generate a 3D object within 9s without the obvious drop in performance.

| Methods | FID↓ | KID(%)↓ | COV(%)↑ | MMD(‰)↓ |
|---|---|---|---|---|
| Rodin (Wang et al., 2022) | 84.55 | 4.5 | 33.79 | 11.47 |
| **DiffTF (Ours)** | **25.36** | **0.8** | **43.57** | **6.64** |

Table 1: Comparison of different 3D-aware modules.

| Methods | Overall Score↑ | Texture Score↑ | Geometry Score↑ |
|---|---|---|---|
| EG3D (Chan et al., 2022) | 8.66 | 11.79 | 10.77 |
| GET3D (Gao et al., 2022) | 20.68 | 22.05 | 22.06 |
| NFD w/ texture (Shue et al., 2022) | 0.48 | 1.02 | 0.51 |
| **DiffTF (Ours)** | 70.18 | 65.64 | 67.17 |

Table 2: User study of top 4 methods on ShapeNet and OmniObject3D.

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

| Methods | FLOPs | Parameters |
|---|---|---|
| NFD w/ texture (Shue et al., 2022) | 250.78 G | 124.01 M |
| DiffRF (Gao et al., 2022) | 778.54 G | 251.29 M |
| **DiffTF (Ours)** | 514.84 G | 734.40 M |

Table 3: Comparison of parameters and FLOPs of diffusion-based methods.

| Category | Training Setting | Method | FID↓ | KID(%)↓ | COV(%)↑ | MMD(‰)↓ |
|---|---|---|---|---|---|---|
| Car | Single-category | DiffTF | 36.68 | 1.6 | 53.25 | 2.57 |
| | Multi-category | DiffTF | 42.40 | 1.9 | 48.27 | 3.07 |
| Plane | Single-category | DiffTF | 14.46 | 0.8 | 45.68 | 2.58 |
| | Multi-category | DiffTF | 19.92 | 1.1 | 41.02 | 2.91 |
| Chair | Single-category | DiffTF | 35.16 | 1.1 | 39.42 | 5.97 |
| | Multi-category | DiffTF | 39.74 | 1.3 | 35.14 | 6.38 |

Table 4: Quantitative comparison of single-category model and multi-category model on the ShapeNet. *Single-category* represents the model only generate 3D objects of one category. *Multi-category* represents the unified generative model for multi-category 3D objects. It proves that our multi-category model can maintain a promising performance without obvious drop in generation performance.

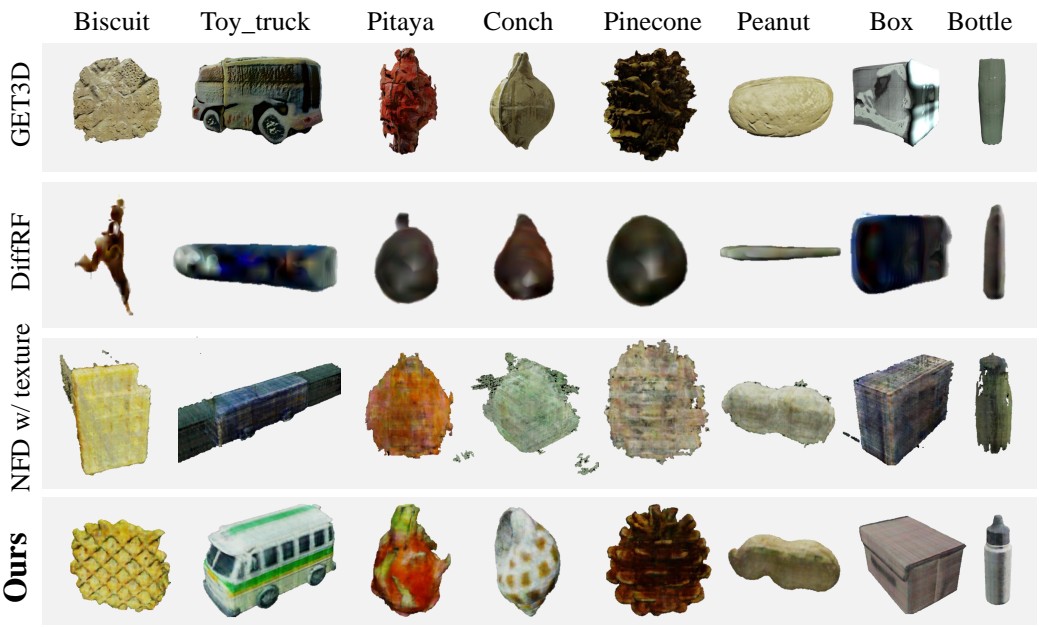

Figure 11: Comparison against other methods on class-conditional generation.

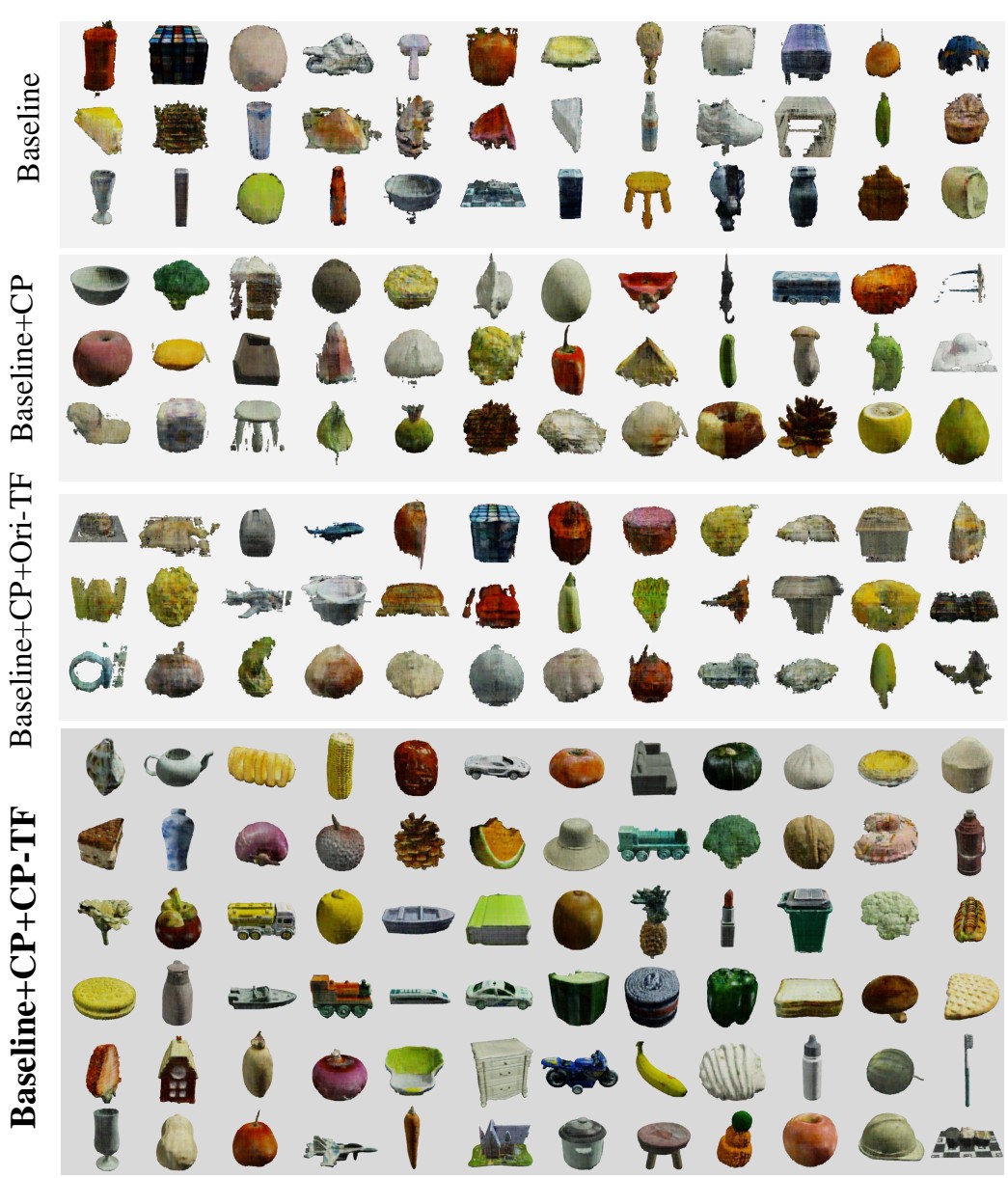

Figure 12: Additional comparisons with different structures.