# OpenReview forum: "Large-Vocabulary 3D Diffusion Model with Transformer"
_ICLR.cc/2024/Conference — ICLR 2024 poster_

### Official Review · Reviewer_veme · 2023-10-30

**Soundness:** 3 good
**Presentation:** 3 good
**Contribution:** 3 good
**Rating:** 6
**Confidence:** 3

**Summary:**

This paper aims at increasing robustness over a wide range of objects for 3D content generation. It uses tri-plane as a compact 3D representation. To fuse the information on three axis, it proposes to use cross attention, named 3D-aware transformer. Experiments on Omniobjects and ShapeNet show that the proposed method produce realistic results.

**Strengths:**

1. The generated 3D objects are realistic. The geometry is better than prior works.
2. Using cross-attention to achieve information fusing of tri-plane is interesting and also efficient in terms of 3D processing.
3. Directly training a 3D diffusion model is hard. The paper is one of the good start.

**Weaknesses:**

1. The paper lacks the introduction of the whole pipeline which links each module together and also the training pipeline which demonstrate which module is pre-trained and which is trained in each step. I am still confused how Step 1 and 2 are related, especially how the fitted tri-plane are used, or is it only the tri-plane decoder is used.
2. The improvement over GAN-based method is not obvious when the texture is on. They all seem realistic.
3. The time complexity should be analyzed, with comparisons over prior methods.

**Questions:**

See Weaknesses.

---

> ### Author Response · Authors · 2023-11-22
>
> [a] **Introduction of our pipeline**. Thanks for your careful advice. Note that in the sampling process, Step 1 aims to generate the triplane via the trained diffusion model while Step 2 focuses on rendering those generated triplanes via the trained triplane decoder. The fitted triplanes are only used to train the diffusion model. The trained decoder will used in fitting the triplanes (training) and rendering the generated triplanes (sampling).
>
> We have revised our paper and added a detailed introduction of our workflow in the Appendix. Section 1.2 as follows:
>
> Training: The training process consists of two steps. In the **step I. triplane fitting**, the objective is to obtain the diverse triplane features and robust triplane decoder. Therefore, for clarification, we divide **step I** into two subtasks: **step I-I** training shared decoder and **step I-II** optimizing triplanes from diverse 3D objects.
> To maintain the robustness of the decoder, we adopt around 20 percent diverse and high-quality objects for optimizing the shared decoder in the **step I-I**.
> Then, in **step I-II**, we adopt the trained decoder with frozen parameters to merely fit the triplanes. Finally, we can use those trained triplanes as the ground truth to train the 3D-aware transformer-based diffusion model in **step II**.
>
> Sampling: Similar to the training process, sampling the 3D content from DiffTF has two individual steps: 1) using a **trained diffusion model** to denoise latent noise into triplane features, and 2) adopting the **trained triplane decoder** to decode the implicit features into the final 3D content.
>
>
> [b] **Comparison with GAN-based methods**. Although GAN-based methods can generate a series of 3D objects, as we have shown in our paper, they are not skilled in generating both the details of objects and rich semantical information. The semantic information of those 3D contents is ambiguous, posing huge challenges in categorizing those objects from a human perspective. Therefore, there is a significant gap between the generated objects of GAN-based methods with real-world objects.
> In contrast to GAN-based approaches, our method can handle the challenges effectively relying on the novel 3D-aware modules. Benefiting from the generalized 3D prior and specialized feature, the 3D objects generated by our method encapsulate rich semantic information and reasonable details, making them more similar to real-life objects.
>
> [c] **Analysis of complexity**. Thanks for your kind suggestion. We have provided the related information in the new version of the Appendix (Table. 3). Without the most efficient structure, our DiffTF is capable of generating a 3D object within 30 seconds. Additionally, to accelerate the speed of sampling further, we can adopt the DDIM with 50 diffusion steps. Adopting this setting, DiffTF can generate a 3D object within 9s.
>
> Methods|FLOPs|Parameters
> ---|---|---
> NFD w/texture |250.78G|124.01M
> DiffRF|778.54G|251.29M
> **DiffTF**|514.84G|734.40M
>
> We sincerely appreciate your great efforts in reviewing this paper. Your constructive advice and valuable comments really help improve our paper. Please let us know if you have follow-up concerns. We sincerely hope you can consider our reply in your assessment, and we can further address unclear explanations and remaining concerns if any.

---

### Official Review · Reviewer_heX9 · 2023-10-31

**Soundness:** 4 excellent
**Presentation:** 4 excellent
**Contribution:** 2 fair
**Rating:** 6
**Confidence:** 4

**Summary:**

The paper presents a 3D generation pipeline based on triplane diffusion. To achieve this, triplane fitting is conducted on the train set, and the output triplanes are used as the inputs to the diffusion model. Initializing as three pure noise planes, the paper uses a transformer-based architecture with cross-plane attention to enhance the 3D-awareness of the generation.

**Strengths:**

* The task of 3D generation, especially on large-scale open vocabulary datasets is challenging and well-motivated.
* The paper is elegantly written and easy to understand.
* Through ablation studies were done to justify its design choices.
* Evaluations were done with reasonable metrics using both older ShapeNet dataset, and new large-scale OmniObject3D. Large margin of improvements can be observed.

**Weaknesses:**

* The contribution of the paper mainly comes from its transformer-based architecture and combining existing tricks in designing such architectures. The overall workflow has been introduced and well studied by prior works such as [1] and [2].
* Upon the efforts of enhancing 3D-awareness of the generation, the rendered results still suffer from noisy geometry and texture details. It is not well-discussed what could be the source of these limitations.



[1] Shue, et al. 3d neural field generation using triplane diffusion, in CVPR, 2023.
[2] Chen, et al. Single-Stage Diffusion NeRF: A Unified Approach to 3D Generation and Reconstruction, in ICCV, 2023.

**Questions:**

None

---

> ### Author Response · Authors · 2023-11-22
>
> [a] **Contribution of DiffTF**. Although the triplane diffusion pipeline has been introduced in prior works, they remain focused on a single category. NFD[1] trains different models for different categories and Rodin [2] only validates in human data. Different from single-category generation, large-vocabulary generation is more challenging because of a) the need for expressive yet efficient 3D representation; b) large diversity in geometry and texture across categories; c) complexity in the appearances of real-world objects.
>
> Furthermore, we conduct comparisons against NFD and Rodin. The experiments strongly prove the superiority of our DiffTF in large-vocabulary 3D generation.
>
> Thus, our main contributions are as follows:
> 1) Our DiffTF is the first attempt to achieve large-vocabulary 3D object generation with a single feed-forward generative model.
> 2) Benefiting from the two 3D-aware modules, DiffTF can extract the generalized 3D prior and specialized features for handling the three special challenges in large-vocabulary generation.
> 3) Compared with other SOTA methods, our DiffTF can generate more realistic and high-quality 3D objects with rich semantical information.
>
> [b] **More discussion on limitation**. We thank you for bringing this issue to our attention. We contend that one potential reason is the unoptimized noise scheduling scheme. According to prior work in image diffusion[1], adopting an appropriate noise scheduling scheme can improve the generative performance. Choosing the most proper noise scheduling scheme can boost the performance of DiffTF further.
>
> [1] Chen T. On the importance of noise scheduling for diffusion models[J]. arXiv preprint arXiv:2301.10972, 2023.
>
> We sincerely appreciate your great efforts in reviewing this paper. Your constructive advice and valuable comments really help improve our paper. Please let us know if you have follow-up concerns. We sincerely hope you can consider our reply in your assessment, and we can further address unclear explanations and remaining concerns if any.

---

> > ### Comment · Reviewer_heX9 · 2023-11-22
> >
> > Many thanks to the authors for their replies, I do not have further questions and will maintain my score.

---

### Official Review · Reviewer_yEnj · 2023-11-01

**Soundness:** 3 good
**Presentation:** 1 poor
**Contribution:** 3 good
**Rating:** 6
**Confidence:** 4

**Summary:**

This paper proposes a two-stage 3D generation method, where 1) in the first stage, triplanes and a shared decoder are fitted on Shapenet/OmniObject 3D dataset. 2) in the second stage, a diffusion model is trained on the fitted triplanes. The author proposes a novel transformer-based denoiser (based on the proposed cross-plane attention) for training the diffusion model, and show state-of-the-art 3D generation results.

**Strengths:**

1. It makes a lot of sense to use attention to model the inter- and cross-plane relations in a triplane, and learn such relations on massive amount of data. Ablation studies show that this is very effective.    2. The proposed method achieves state-of-the-art generation results on both Shapenet and OmniObject3D datasets in terms of geometry and texture quality, and diversity.

**Weaknesses:**

1. I feel that this paper’s writing is very confusing, and could be improved significantly for clarity.
    - The method section shows that this is a two-stage model; but Fig. 2 seems to suggest that this is a single-stage model. The authors might want to make Fig. 2 clearer.
    - For figure 4, since the generation results on OmniObject3D are class-conditioned, would it make sense to arrange the results in a specific order of classes in order for the readers to easily compare different methods?
    - I have to dig into the appendix to find the architecture of the shared triplane decoder; there’s no reference to appendix in section 3.1. What’s worse, even looking at the appendix, I’m still unable to find out the width of the MLP layers for the triplane decoder.
    - I find it a bit hard to imagine what exactly the architectures are for the different variants in the ablation “Studies of 3D-aware transformer modules”. The authors might want to elaborate more in the appendix, or better draw figures to the ablated architectures.
    - Section 4.3 mentioned about the importance of triplane normalization. But this is never mentioned in the method section. In fact, the authors might want to show a histogram of the unnormalized triplane values in the appendix.

**Questions:**

1. What is the prediction objective of DiffTF’s diffusion model? Is it \epsilon-prediction, x_0 prediction or v-prediction?
2. What’s the size of the shared triplane decoder?
3. For the 3D-aware transformer part, as the triplane resolution is only 16x16 and the patch size is 2, this means that there’re only 8x8=64 tokens on each plane, and 64x3=192 tokens in total. Why not just do self-attention over all these 192 tokens, instead of using the cross-plane attention? Would this simpler architecture also work? In fact, looking at figure 3(b), such simple self-attention is already used in the 3D-aware encoder/decoder.
4. How does the proposed cross-plane attention compared against Rodin’s 3D-aware convolution? Does it make sense to compare with this work qualitatively and quantitatively, as the cross-plane attention seems a major contribution of this work?
	Rodin: A Generative Model for Sculpting 3D Digital Avatars Using Diffusion
5. It seems that the model is trained per-category on shapenet cars, chairs, airplanes? Why not train a class-conditioned generator just like what has been done on the OmniObject3d data?

---

> ### Author Response · Authors · 2023-11-22
>
> [a] **Workflow of DiffTF**. Thanks for your careful advice. We have revised the Fig. 2 and the caption to make it more legible. The top of the new figure represents the first step while the bottom represents the second one.
>
> [b] **Comparison in a specific order**. Thanks for bringing this issue to our attention. According to your suggestion, we have added the additional visualization involving all class-conditional generative methods in the Appendix (Fig. 11). Note that EG3D does not support the class-conditional generation.
>
> [c] **Details of the share-weight decoder**. We thank you for your kind advice. We have added the reference to the Appendix in Section 3.1. Detailed information about our triplane decoder has been added in the Appendix (Fig.2). The size of MLP can also be found in this figure. Please refer to it.
>
> [d] **Architectures in ablation studies**. We appreciate your detailed suggestion. For clarification, we have included a figure to illustrate the different structures used in our ablation studies in Appendix (Fig. 7)
>
> [e] **Distribution of fitted triplanes**. Thank you for your insightful comment. We have added a histogram in the Appendix (Fig.13) to show the distribution of triplane values. As we mentioned in our paper, the range of -10 to 10 is too large for optimizing the diffusion model. Therefore, we normalize the triplane values to the range of -1 to 1.
>
> [f] **Prediction objective of DiffTF**. In the Appendix. Section 3.1, we report that we use \epsilon as the prediction objective of DiffTF.
>
> [g] **Cross-plane attention in DiffTF**. In this paper, we try to extract the generalized 3D prior from diverse objects for raising the performance of large-vocabulary generation. Therefore, the novel cross-plane attention should focus on the interdependencies among planes rather than the specialized feature mainly introduced by self-attention.
>
> Furthermore, the comparison of Ori-TF and CP-TF in our ablation studies (Table. 3) proves that the combination of cross-plane attention and self-attention (FID: 25.36) is better than the pure self-attention modules (FID: 52.35).
>
> [h] **Comparison with Rodin**. We appreciate your valuable advice. We have added the comparison with Rodin in the Appendix (Table. 1). As shown in the following table, our 3D-aware models achieve an impressive improvement in generative performance compared with 3D-aware convolution in Rodin.
>
> Methods|FID|KID(%)|COV(%)|MMD(‰)|
> ---|---|---|---|---
> Rodin|84.55|4.5|33.79|11.47
> **DiffTF**|**25.36**|**0.8**|**43.57**|**6.64**
>
>
> [i] **Class-conditional generation on ShapeNet**. Note that most prior works, such as NFD, evaluate the generative methods on a single category. For fairness of evaluation, we adopt the same setting.
> Furthermore, we have added the comparison of the class-conditional model trained on three categories of ShapeNet in the Appendix (Table. 4). The comparison between our single-category model and multi-categories model in the following Table shows that our method can extend to multi-categories without bringing significant drop in performance.
>
> Category|Training Setting|Method|FID|KID(%)|COV(%)|MMD(‰)
> ---|---|---|---|---|---|---
> Car|Single-category|DiffTF|36.68|1.6|53.25|2.57
> Car|Multi-category|DiffTF|42.40|1.9|48.27|3.07
> Plane|Single-category|DiffTF|14.46|0.8|45.68|2.58
> Plane|Multi-category|DiffTF|19.92|1.1|41.02|2.91
> Chair|Single-category|DiffTF|35.16|1.1|39.42|5.97
> Chair|Multi-category|DiffTF|39.74|1.3|35.14|6.38
>
>
> We sincerely appreciate your great efforts in reviewing this paper. Your constructive advice and valuable comments really help improve our paper. Please let us know if you have follow-up concerns. We sincerely hope you can consider our reply in your assessment, and we can further address unclear explanations and remaining concerns if any.

---

> > ### Comment · Reviewer_yEnj · 2023-11-22
> >
> > Really appreciate your response. I'm maintaining my score.

---

### Author Response · Authors · 2023-11-22

We sincerely appreciate all reviewers for their insightful suggestions and careful advice on our work. We are glad that the reviewers contend that our method “**is very effective**” and “achieves **state-of-the-art** generation results” (Reviewer yEnj); “The paper is **elegantly written and easy to understand**” and “**Large margin of improvements** can be observed” (Reviewer heX9); “The paper is one of the **good start**” and “Using cross-attention is **interesting and also efficient**” (Reviewer veme). We appreciate them and carry them to our future work.

We have posted responses to the comments of each reviewer, hoping our response can address your concerns.

We have also uploaded the revised paper and supplementary materials (the changes are presented in blue color). The modifications are mainly summarized as follows:

1.	We further compare DiffTF with Rodin in the Appendix (Table. 1)
2.	We revised the Fig. 2 in our main paper to make it more clear.
3.	We add more visualization results on OmniObject3D in the Appendix (Fig. 11).
4.	We add more details about the share-weight decoder in the Appendix (Fig.2).
5.	We include the comparison of structures used in ablations in the Appendix (Fig. 7).
6.	We report the detailed distributional information of triplanes in the Appendix (Fig.13)
7.	We add the class-conditional results of DiffTF on ShapeNet in the Appendix (Table. 4)
8.	We extend the detailed workflow of DiffTF in the Appendix. Section 1.2
9.	We analyze the complexity of diffusion-based methods in the Appendix (Table. 3)

Finally, we sincerely thank our reviewers and look forward to further discussions with you.

---

### Meta-Review · Area_Chair_WCnd · 2023-12-12

**Metareview:**

The paper has received three borderline accept recommendations with scores of 6/6/6. Reviewers collectively acknowledge the paper's strength in enhancing 3D diffusion models with transformer architecture, enabling the generation of a large vocabulary of object categories.

However, reviewer heX9 raised concerns about the novelty of the work compared to existing research. After reviewing the rebuttal, the Area Chair (AC) concurred with the observation that the paper mainly introduces transformer-based architectures compared to existing work, while the overall workflow aligns with existing approaches.

Upon careful examination of the manuscript, the AC identified grammar issues and typos, such as "pre-train" needing correction to "pre-trained" in the related work section. The AC suggests that improving the quality of the manuscript is necessary.

In summary, the paper introduces a transformer-based 3D diffusion model based on tri-plane representations, contributing to the advancement of 3D generative modeling. While there are concerns about novelty and manuscript quality, all reviewers acknowledge the paper's contribution. The AC does not find enough evidence to overturn the decision and recommends acceptance with the suggestion to enhance the manuscript's quality.

**Justification For Why Not Higher Score:**

The novelty of the paper is the major bottleneck for getting a higher score. The overall workflow is similar to existing 3D diffusion model approaches. Also, it misses the opportunity to broaden its impact by comparing with existing text-to-3D generative modeling works, which are based on single-view or multi-view diffusion models and has already been able to generate 3D assets with open-vocabulary inputs.

**Justification For Why Not Lower Score:**

The paper still contributes to 3D generative modeling directly on 3D representations. All reviewers agree on the contribution and the AC does not find enough evidence to overturn the decision.

---

### Decision · Program_Chairs · 2024-01-16

Accept (poster)